# Duplicate Genes Contribute to Variability in Abiotic Stress Resistance in Allopolyploid Wheat

**DOI:** 10.3390/plants12132465

**Published:** 2023-06-28

**Authors:** Linying Du, Zhenbing Ma, Hude Mao

**Affiliations:** 1State Key Laboratory of Crop Stress Biology for Arid Areas, College of Life Science, Northwest A&F University, Yangling 712100, China; dulinying1201@163.com (L.D.); mazhenbing1@163.com (Z.M.); 2State Key Laboratory of Crop Stress Biology for Arid Areas, College of Agronomy, Northwest A&F University, Yangling 712100, China

**Keywords:** gene duplication, abiotic stress, wheat, evolution, functional divergence

## Abstract

Gene duplication is a universal biological phenomenon that drives genomic variation and diversity, plays a crucial role in plant evolution, and contributes to innovations in genetic engineering and crop development. Duplicated genes participate in the emergence of novel functionality, such as adaptability to new or more severe abiotic stress resistance. Future crop research will benefit from advanced, mechanistic understanding of the effects of gene duplication, especially in the development and deployment of high-performance, stress-resistant, elite wheat lines. In this review, we summarize the current knowledge of gene duplication in wheat, including the principle of gene duplication and its effects on gene function, the diversity of duplicated genes, and how they have functionally diverged. Then, we discuss how duplicated genes contribute to abiotic stress response and the mechanisms of duplication. Finally, we have a future prospects section that discusses the direction of future efforts in the short term regarding the elucidation of replication and retention mechanisms of repetitive genes related to abiotic stress response in wheat, excellent gene function research, and practical applications.

## 1. Introduction

Diversity and phenotypic variation in plants do not arise exclusively from the emergence of completely novel and diverse genes; in many plants, diversity may instead arise primarily through gene duplication and adaptive specialization of preexisting genes. Gene duplication is an evolutionary process in which genetic diversity and new functions are generated via whole-genome duplication (WGD) events or smaller-scale, single-gene or single-base duplications [1,2], the occurrence of which in a gene results in two genes that cannot be functionally distinguished from each other. In the evolutionary history of plants, animals, and fungi, gene duplication events have occurred ubiquitously across kingdoms [3,4,5,6]. In comparison with other eukaryotic genomes, plants typically have a higher rate of evolution, which entails continuous increases in their genomic diversity [7,8]. Recently, previous whole-genome duplication events that led to novel functions, such as biotic or abiotic stress tolerance, higher grain weight and quality, or phenotypic changes in flowering time or plant height, were detected in several domesticated crops, including wheat (*Triticum aestivum*), cotton (*Gossypium hirsutum*), and soybean (*Glycine max*).

If a gene duplication results in two copies with a sufficiently similar nucleotide sequence, then these two genes should also share functional overlap, and considerable research efforts have focused on examining the evolutionary mechanisms through which both genes are retained. Currently, there are several models that could explain the retention of duplicated genes. In eukaryotes, most duplicated genes that are retained become non-functional due to disruption or loss-of-function mutations in regulatory elements; alternatively, ancestral functions are sometimes partitioned among the duplicated genes. However, neither case leads to the evolution of new functions, and this process is thus termed “nonfunctionalization” [9]. In addition, duplicated gene retention is controlled by other evolutionary mechanisms, such as neofunctionalization, functional specialization by sub-functionalization, or dosage amplification [9]. Interestingly, despite the different effects of these various evolutionary mechanisms, duplicated genes can still retain some level of functional redundancy [10], which can increase the plasticity of a genome or enhance the adaptability of a species to changing environments [11].

Plants cannot escape from stress-inducing environmental conditions through movement and, thus, the strong selection pressure imposed by these conditions increases the likelihood and frequency of gaining or losing stress-responsive genes. In varying stress conditions, stress-sensing and stress-response mechanisms should be able to evolve rapidly as needed to adjust to new stimuli and therefore require continual innovation in their genetic basis. Duplicate genes have been proposed to serve as the main source of evolutionary novelty and a possible source of functional innovation. Harsh weather events or climatic conditions are becoming increasingly frequent due to global climate change and the world now faces the prospect of food insecurity.

Bread wheat (AABBDD) is one of the three major staple crops, globally, and future wheat yields will be critical to human survival. Allohexaploid wheat contains three related subgenomes, presumably with triplicate copies of the large majority of genes, resulting in highly complicated gene interaction networks due to the presence of these multiple genomes in each cell [12,13,14]. Gene and genome duplications have likely provided significant contributions to the morphological complexity of plants [15], and these duplication events may also influence the physiological complexity of stress responses. For example, the presence or absence of *HPT2* (a low-temperature-responsive gene in wild barley) in the pangenome of barley (diploid), wild emmer (tetraploid), and bread wheat (hexaploid) lines appears to be related to gene dosage constraint and environmental adaptation [16]. Evidence in previous studies at least partially supports that environmental stresses, such as drought, cold, heat, and salt, contribute to the retention of duplicated genes [17,18,19]. Herein, we review recent advances in our understanding of the role of duplicate genes in plant resistance to abiotic stress.

## 2. Diversity and Divergence of Duplicate Genes Involved in Abiotic Stress Resistance

Distinct from most other eukaryotes, plants cannot move to avoid unfavorable conditions, and thus their genomes tend to evolve at faster rates under these selective conditions, resulting in higher genomic diversity that may enable survival in stressful environments [7]. Plant genomes typically contain a high number of duplicate genes, with 65% of annotated genes, on average, having a duplicate copy in the genome [17]. Several genomic sequencing analyses in plants suggest that whole-genome duplication has occurred several times during the past 200 million years of angiosperm evolution [20,21,22]. In other eukaryotes, such as humans and budding yeast, the most recent whole-genome duplication events occurred roughly 450 million years ago (Mya) and 200 Mya, respectively [6,23,24]. On the other hand, some plant species include both diploid and polyploid individuals [25]. In those species, duplicated genes can increase genetic variability, contributing to increased complexity. spatio-temporal transcriptomic plasticity, and the higher adaptability of polyploids to environmental stress [26,27].

Wheat genomes are relatively large compared with other major cereal crops; diploid einkorn wheat is ~5.0 gigabases (Gb) [28,29], tetraploid emmer is >10 Gb [30], and exaploidy bread wheat is ~17 Gb [31]. Despite the complex composition of the wheat genome, more than 80% of each genome is made up of repetitive DNA sequences [31]. In addition, these large, heterozygous exaploidy wheat genomes contain a remarkable diversity of genetic variations, providing a background conducive to gene duplication, functional/phenotypic variation, and evolutionary conservation. Diversity in duplicated genes is inextricably linked to differences in the wheat genome and differences in genetic diversity among subgenomes [32,33]. Although a majority of wheat genes have three copies distributed across the A, B, and D subgenomes, studies have shown that the degree to which vegetative traits are inherited differs among wheat subgenomes, with trans-acting variants, for example, being more genetically diverse in genomes A and B than in D [34,35,36,37]. This variability in the inheritance of different gene copies suggests that genomic interactions could play an important role in regulating genes selected for domestication and improvement of agricultural traits and stress resistance in wheat.

Receptor-like protein kinases (RLKs), the largest gene family in plants, play critical roles in the regulation of plant developmental processes, signaling transmission, and stress resistance [38,39,40]. In recent years, the RLK family of wheat has been identified, and collinearity events and tandem gene clusters results suggested that polyploidization and tandem duplication events contributed to the RLK member expansions of *T. aestivum* [38]. Among them are lectin receptor kinases (LEC-RLK) [41], leucine-rich repeat receptor-like kinases (LRR-RLKs) [42,43], Cysteine-rich receptor-like kinases (CRKs) [44], thaumatin-like proteins and thaumatin-like kinases (TLPs) [45], and proline-rich extensin-like receptor protein kinases (PERKs) [46,47]. Gene replication events were identified in these families, and some of these replication gene pairs had differential expression data under abiotic stress, such as *TaTLP14-A1/B1/B5* being up-regulated in expression under osmotic stress, while *TaTLP14-B4* exhibited the opposite [45]. Enzymatic antioxidants are an important stress-responsive class of proteins that scavenge excess reactive oxygen species (ROS) in the presence of cofactors such as copper and zinc ions [48]. Several studies identified and reported duplication in antioxidant genes, including catalase (CAT) [49,50], superoxide dismutase (SOD) [51,52,53], ascorbate peroxidase (APX) [54], glutathione peroxidase (GPX) [55,56], peroxidase (POD) [57], and glutathione reductase (GR) genes in bread wheat [58]. For example, *TaCAT3-A1* and *TaCAT3-A2* were found to be clustered into tandem duplication event regions, while the number of *cis*-elements in the promoter of *TaCAT3-A2* was more than *TaCAT3-A1*; moreover, *TaCAT3-A1/A2* contained *cis*-elements associated with cold response, but not exist in other subgenome copies *TaCAT3-B* and *TaCAT3-D* [50], which suggests that there may be differences in the cold response of *TaCAT3* homologous gene groups.

In plants, various transporters were reported related to the transport of Ca, Na, and other important molecules during stress response to maintain ion homeostasis in the plant cell [59,60,61]. Hyperosmolality-gated calcium-permeable channels (OSCAs) [62,63], boron transporters (BOR) [64,65], mechanosensitive channels of small conductance-like (MSL) genes [66], Ca^2+^/cation antiporters (CaCAs) [67], cation proton antiporters (CPAs) [68], P-type II Ca^2+^ ATPases [69], and thaumatin-like protein kinases (TLPKs) in the bread wheat gene family were identified and analyzed for their expression pattern under abiotic stress [70]. For example, the gene pairs *TaMSL4-A1* and *TaMSL7-A* in MSL were both drought-induced, but the expression of *TaMSL4-A1* was inhibited under heat stress and in the early stage of salt stress, while the expression of *TaMSL7-A* was up-regulated under heat stress and the under early salt stress, which indicates the response of paraline homologous genes was different under different abiotic stresses; the expression was also diverse and began to express and function under different levels of stress [63]. The cation proton antiporter (CPA) superfamily, including K^+^ efflux antiporter (KEA) and cation/H^+^ exchanger (CHX) family proteins as well as the Na^+^/H^+^ exchanger (NHX), *TaNHX4-B.1,* and *TaNHX4-B.4,* facilitated differential drought, salt, and heat stress tolerance to *Escherichia coli* [64]. A similar phenomenon occurs with transcription factors, in which orthologous or paralogue genes that are produced via gene duplication exhibit identical or opposite expression patterns when subjected to abiotic stress, including *NAC* [71,72,73,74], *DREB* [75], *Hsf* [76,77,78,79], *MYB* [80,81], *bZIP* [82,83], *WRKY* [84,85], *AP2/ERF* [86], *GRF* [87], and the homeobox genes *HD-Zip* [88], *TALE* [89,90], *ZF-HD* [91], and *WOX* [92,93]. Taking *TaTAIL* as an example, Rathour et al. systemically identified and analyzed *TAIL* family members in wheat, including gene and protein structural properties, phylogeny, and expression patterns. Gene duplication events were identified, including gene pairs *TaTALE8-4A3* and *TaTALE8-4A1* produced via fragment replication and five tandem duplicate gene pairs such as *TaTALE1-1A2* and *TaTALE1-1A1*. The *cis*-acting elements of these repeating gene promoters and their expression data were different under heat and salt stress, indicating that the response of repeating genes to abiotic stress was diverse [89].

The Introduction of diversity in duplicate genes can be grouped into two main categories based on the type of alterations in DNA, that is, changes in gene structure or epigenetic modifications. Changes in DNA sequence mainly impact diversity and variability at the transcriptional and post-transcriptional levels, as well as in the translated protein or post-translational modifications [94,95,96,97,98,99]. At the transcriptional level, differences in *cis*-acting elements can result in differences in response to the same stimulus between homologous genes, while changes in promoter region binding sites for trans-acting factors can trigger differences in expression between gene copies [34]. Previous research has shown that stress-responsive plant genes are retained at higher rates than nonfunctionalized duplicate genes, especially transcription factors and signal transduction proteins. For instance, the expression levels of 7 out of 25 *TaADF* genes (*TaADF13*/*16*/*17*/*18*/*20*/*21*/*22*) were significantly affected by cold or freezing treatment, while overexpression of *TaADF16* enhanced tolerance to freezing in wheat plants [100]. Additionally, a number of other transcription factor genes have been reported to respond to and regulate drought stress tolerance, including *TaOPF29a*, *TaDrAp1/2*, *TaFDL2-1A*, *TaSNAC4-3D*, *TaMpc1-D4*, *TaGT2L1D*, *TaSNAC4-3A*, *TaWRKY1-2D*, *TaNFYC-A7*, and *TaERF-6-3A* [73,74,85,86,101,102,103,104,105]. Besides, genome-wide association studies (GWAS) have identified sequence variations in homologous genes that could increase drought tolerance in wheat, such as *TaNAC071-A*, *TaDTG6-B*, and *TaSNAC8-A* [71,72,75].

Increased diversity and divergence of duplicated genes is also linked to neofunctionalization or sub-functionalization in abiotic stress response, such as heat stress transcription factors (HSFs). HSFs are among the most important TFs in plant response to heat stress, but some HSFs also respond to drought or salt stress. For example, overexpressing *TaHsaA2d* or *TaHsaA6f* in wheat not only increases heat tolerance but also drought and salt stress, respectively [76,106], while overexpression of *TaMYB344, TaAIDFa*, or *TaAREB3* enhances tolerance to drought, heat, or salt stress in transgenic lines [80,107,108]. In post-transcriptional regulation, changes in the DNA sequence can introduce variable shear events, such as splice site disruption, which can lead to changes in target gene transcripts that consequently affect the structure and function of the translated protein [98,109,110]. These results illustrate how the accumulation of homoeologs with biased expression patterns can affect stress tolerance. In particular, changes in the DNA sequence between homologous TFs may impact their DNA binding and transcriptional regulatory activities, such as a *TaDTG6-B* gain-of-function allele that improves drought tolerance in wheat [75].

Changes in the epigenetic modification of duplicated DNA sequences induced by environmental stimuli can also affect heritable variation in gene expression [111,112,113]; such modifications include DNA methylation, histone methylation and acetylation, or modifications to mRNAs or non-coding RNAs [114,115,116,117,118,119]. It should also be noted that the epigenetic modification landscape across the three subgenomes of hexaploid wheat may contribute the predominant regulation to gene dosage for some genes. For example, DNA methylation and acetylation modifications modulate the expression of *TaCYP81D5*, which contributes to both seedling- and reproductive-stage salt tolerance in bread wheat [120].

The combination of these factors, along with the complexity of the wheat genome, provides rich potential for variation among duplicated genes, from the DNA to protein levels, especially in TFs that function as the major regulators of abiotic and biotic stress response [121]. These altered TF homoeologs can thus participate in sophisticated and versatile regulatory networks that facilitate adaptability and maintenance of homeostasis for essential biological processes in wheat in the face of highly variable and extreme climatic conditions. Genetic diversity causes structural and functional phenotypic differences, and diversity among duplicated genes considerably enriches the genetic diversity of the wheat genome. The resulting complex regulatory networks can further improve the plasticity of signal transduction processes to some extent, and also provide resources for environmental adaptation in both natural evolution and artificial selection processes.

## 3. Contribution of Duplicate Genes to Abiotic Stress Resistance

Transcriptional regulation underlies all biological activities in plants, and especially provides a sophisticated set of mechanisms for reacting to changes in the external environment. Duplicated genes typically exhibit distinct patterns of expression, and this regulatory divergence has been proposed to serve as the prelude to functional differentiation among duplicate genes. Thus, duplicate genes are a major source of potential functional innovations in plant response mechanisms to abiotic stress stimuli [120,122,123,124]. Differentiation in expression patterns among duplicated genes is considered a precursor to functional differentiation of genes, because functional divergence may occur long after duplication, whereas changes in expression patterns might begin relatively soon (or immediately) after the duplication event [125]. Although duplicated genes (i.e., homologs and paralogs) may share a high degree of structural or functional similarity, differences in their response to environmental stress can be related to variations in *cis*-acting regions [126], transcription factor binding sites, and/or methylation status [127]. In particular, the gain of additional or different *cis*-regulatory elements in the promoter regions of MIKC-type MADS-box [128]; CHY zinc-finger and RING finger [129]; basic leucine zipper [130]; abscisic acid-, stress- and ripening-induced (ASR) [131]; APETALA2/Ethylene-Responsive Factor (AP2/ERF) [132,133]; E3 ubiquitin ligase [134]; or LATERAL ORGAN BOUNDARIES DOMAIN (LBD) [135] genes results in enhanced abiotic stress tolerance in different crops.

The evolution of TATA boxes in duplicated genes may also provide some clues regarding the interrelationship of environmental stress, divergent expression patterns, and the conservation of duplicated genes [136]. Moreover, de novo functionalization is closely correlated with the retention of duplicated genes that inherit little or none of the original function [137]. For instance, *CPK7* and *CPK12*, which are duplicated wheat genes that together comprise the *CDPK* family, are located on chromosomes 2B and 5A, respectively. The promoters of *CPK7* and *CPK12* have different *cis*-acting elements and their expression levels vary. Notably, *TaCPK7* is expressed in response to drought, salt, cold, and hydrogen peroxide, whereas *TaCPK12* is only expressed in response to ABA treatment. This functional divergence has complementary or amplifying effects on the *TaCPK7*-dependent stress signaling network in wheat [138,139].

Previous studies have demonstrated that *cis*-acting elements or trans-acting factors can also cause differences in the expression level of duplicated genes [34]. For instance, heat shock transcription factors (HSFs) upregulate the transcription of target-gene-encoding heat shock proteins (HSPs) via recognition and binding to promoter region heat shock elements (HSEs) in response to heat stress [77,140]. Atypical HSEs, containing mismatched nucleotides at specific positions, are more sensitive/responsive to heat stress and are expressed at higher levels than typical HSEs [96]. Thus, the distribution of diverse HSE motifs in the promoter regions of duplicate HSP genes (homologous or tandem repeats), combined with the effects of the HSE sequence variation, make the *TaHSF*-*TaHSP* module essential for heat stress regulation and adaptation in wheat [141]. In addition, similar studies have reported finding homolog expression bias between different copies of the same gene and functional differences due to specific variants, such as *CLPB* (Caseinolytic Protease B), *SKP* (S-phase kinase-associated protein 1), *ALDH* (Aldehyde dehydrogenase), *NAC* (NAM/ATAF1/2/CUC2), and *SOS1* (Salt Overly Sensitive 1) in wheat [142,143,144,145,146,147]. These above results imply that differential regulation of duplicated genes can augment a plant’s ability to respond to environmental abiotic stress challenges.

Phenotypic and functional divergence is facilitated by changes in protein coding sequence between duplicated gene pairs. During gene duplication, structural alterations in regulatory areas, such as protein coding regions and small RNA binding sites, may also occur. Furthermore, gain or loss mutations in exons, introns, pseudo-exons, or indels have been shown to occur more frequently in duplicated genes than in single-copy homologs [148]. Numerous studies have reported finding single-nucleotide alterations or base insertions and deletions that alter the stress response of homologous genes [149,150,151,152]. For example, GWAS identified a favorable allele of *TaDTG6, TaDTG6-B* ^Del574^, that harbored a frameshift mutation due to deletion of a 26-bp DNA fragment in the coding region. Strong protein and DNA sequence interaction properties of the encoded DREB protein allowed it to bind to the DRE/CRT *cis*-acting element, upregulate downstream gene expression, and ultimately enhance wheat drought tolerance [75]. Similarly, a variation in the region of subgenome A, but not subgenome B, near *TaSnRK2.8* and a single-nucleotide polymorphism (SNP) in the *TaSnRK2.8-A-C* region conferred a stronger drought-tolerant response in wheat, accompanied by significantly greater seedling biomass and water-soluble carbohydrate contents [150]. Another recent study revealed that *TaWD40-4B.1*, located in the main drought tolerance QTL, qDT4B, along with an early termination codon generated through a nonsense mutation in *TaWD40-4B*, was highly correlated with drought resistance in a natural wheat population. These findings showed that wheat carrying *TaWD40-4B.1^C^* had significantly higher drought resistance than wheat carrying *TaWD40-4B.1^T^* [151]. In addition, microRNAs (miRNAs) also facilitate the control of gene expression in plant species primarily through detecting and cleaving particular regions on target genes to modify their function. Changes in miRNA binding sites and miRNA precursors of duplicated genes may also have different expression patterns and functions [153,154,155,156].

Gene duplication can play an essential role in preserving the integrity of a genetic system while also mitigating the effects of the surrounding environment on that system. That is, if one copy is inactivated through a mutation, other copies can still perform the original function to compensate for potential damage due to the inactivation [157,158]. This functional redundancy is commonly found in polyploid plants, and research has shown that most partially homologous gene copies in wheat are co-expressed. Multiple studies indicate that partially homologous gene copies can be directed by the same regulatory network [32,158,159], and that this functional redundancy between homologous genes is greatly expanded in heteropolyploid wheat. In one typical example, all five of the *TaCYP81D* tandem repeat genes, generated via lateral doubling within the same subgenome, were experimentally linked to salt tolerance in wheat. Among them, *TaCYP81D5* was shown to potentially influence the ZAT12-mediated ROS signaling pathway in wheat [120]. By contrast, tetraploid wheat expressing the *cyp81d5-aabb* mutant showed no obvious difference from the wild type in salt tolerance, although *CYP81D2* and *CYP81D4* expression was significantly higher in the mutant than in the wild type. These results implied that the presence of other copies in this gene cluster could make up for the absence of one copy. This functional redundancy enables wheat plants to better withstand unfavorable natural genetic variations and, thus, maintain salt tolerance [112].

The above data show that duplicated genes serve as a crucial source for the development of new defenses against abiotic challenges in wheat, and that these defenses let plants continue to evolve in response to environmental stresses that might have been too severe for their ancestors. In addition to the above cases, a list of duplicate genes is provided in Table 1, including homologous and paralogous genes with structural and functional differences that contribute to wheat response to abiotic stress. A few such examples include drought-responsive *TaHVA1*, *TaRAV*, and *TaNAC* [72,73,160,161]; salt-responsive *TaCHYR*, *TaWRKY75*, and *TaKNOX11* [84,90,129]; cold-responsive *TaEXPA, TaICE*, and *TaAREB* [108,162,163]; and heat-responsive *TaMYB, TabZIP*, and *TaHAG* [80,82,83,164,165].

## 4. Molecular Mechanisms of Abiotic Stress Resistance by Duplicated Genes

Duplicate genes are produced through a variety of mechanisms (Figure 1), and the rapid doubling of a genome during polyploidization can trigger large-scale genomic alterations, such as chromosomal rearrangements, gene inversions, and gene loss, in addition to generating a large number of duplicate genes. Other work suggests that polyploidy may be correlated with enhanced stress tolerance and higher reproductive fitness under stress conditions based on evidence that polyploids typically have a broader geographic distribution than diploid relatives [196,197,198,199,200]. Furthermore, crops that have undergone polyploidization are more prevalently cultivated. For instance, the tetraploid wild emmer wheat that is grown today, *Triticum turgidum* ssp. *dicoccoides* (2n = 4x = 28, BBAA), originated from the diploid wheat, *Triticum urartu* (2n = 2x = 14, AA), and its close relative, *Aegilops speltoides* (2n = 2x = 14, BB). *Triticum aestivum* L (2n = 6x = 42, BBAADD) was created through crossing wild emmer wheat with the diploid, *Aegilops tauschii* (2n = 2x = 14, DD) [201]. In the complete published genome of wheat cultivar, Chinese Spring, three copies can be found for more than half of the genes, and the copies are evenly distributed among the three homologous A, B, and D subgenomes [31]. These duplicated genes are referred to as homoeologs, and they share a high degree of sequence similarity as well as functional conservation and redundancy [31].

Although most of these homologous genes show synergistic expression patterns in the wheat population, a few homologous genes show negatively correlated patterns of expression. This inverse relationship is potentially due to regulatory effects of genetic variations, indicated by a gradual shift in the expression profiles of the same homologs from a positive to a negative correlation concomitant with an increasing number of SNPs in the region adjacent to one of the homologs [34]. Other research has shown that typical allohexaploid species have higher salt tolerance than their tetraploid wheat progenitors [164]. Thus, WGDs appear to improve the organismal capacity for adaptation to environmental challenges through introducing new genetic features and increasingly complex intragenomic network interactions [202,203]. Some recent work has uncovered another duplication phenomenon, distinct from WGD, related to significant genomic enrichment with TEs (transposable elements) [204,205,206,207,208], and there are important distinctions in functional enrichment and retention between genes produced via recent duplication and those produced via WGD [205,207]. In addition to WGD events, the pool of duplicate genes has also been considerably increased by numerous, small-scale subgenomic duplication events, including tandem duplications [1], segmental duplications [209], DNA-based transposition [208], and retrotransposon-mediated duplications [209]. Importantly, in each whole-genome or partial-chromosome-segment duplication event, a portion of genes are eliminated while another genomic fragment carrying duplicated genes is preserved to participate in further evolution.

The ability of crops to adapt to severe climatic conditions can depend heavily on copy number variations (CNVs). The *FR-2* locus (*Frost Resistance-2*) has been linked to cold tolerance, which is consistent with a set of *CBF*s (C-repeat binding factors) that were found to regulate pathways involved in cold-climate domestication and cold tolerance [210,211]. Variability in *CBF* gene expression due to CNVs in the corresponding locus have been correlated with a cold-tolerant phenotype [210,211,212,213,214,215]. In particular, differences in gene coding sequence have been identified that directly impact phenotype, such as sequence variations in the CBF12 binding domains between winter and spring wheat accessions that modulate CBF12 binding activity at target loci [216] and enhanced cold tolerance in winter cereals with a high *CBF* copy number compared to that in single-copy spring cereals. In addition, *CBF13* appears to have undergone pseudogenization in spring barley, based on the prevalence of sporadic nonsense codons, whereas its coding sequence remains intact in winter barley [212]. Other cold tolerance loci have been identified in wheat that are also functionally linked to CNVs in *CBF* genes [213,217].

Transposable elements (TEs) are small sequences, typically of viral origin, that mobilize to random locations throughout the genome in a “cut-and-paste” fashion (i.e., excised and reinserted), in the case of DNA transposons, or in a “copy-and-paste” fashion (i.e., RNAs encoding these elements are reverse transcribed and integrated into new sites, leaving the original), in the case of retrotransposons. Duplication can occur if a gene is co-replicated or co-transcribed/reverse transcribed with a TE, then integrated back into the genome at a new site, sometimes disrupting genes at the integration site [218]. Transposons thus mediate two main genomic effects: (1) they mobilize randomly throughout the genome, disrupting genes; and (2) they generate copies of the sequence in proximity to their integration site, resulting in new genes, pseudogenes, and *cis*-regulatory elements. Alterations in environmental conditions such as temperature, light, and water availability can impact transposon activation, in addition to internal factors that contribute to genomic instability. In a recent study examining the genomes of eleven crops, transposons accounted for 22% to 85% of the total genome content [219]. The wheat genome contains a multitude of TEs, which comprise ~85% of the sequence. For example, TE insertions in the promoter regions of *Vrn1* homologs resulted in several loss-of-function mutations related to cold domestication [220]. Thus, the discovery, characterization, and use of TEs in crops, especially wheat, which necessarily entails an in-depth understanding of their role in gene duplication and the functions of the duplicated genes, could serve as a promising direction for future crop breeding.

## 5. Prospects

Gene duplications are mainly produced via polyploidization and WGD events [160,163]. In cereal crops, polyploidization increases the phenotypic diversity of polyploid species, improves the fitness of fixed hybrids, and enriches genetic diversity through altering genome structure [221,222]. Crops that can adapt to new habitats are needed to ensure stable, long-term food production as soil and climate conditions deteriorate globally. Hence, a key strategy for removing genetic barriers to crop improvement is to engineer the polyploid evolutionary process through utilizing distant crosses between related species [223,224]. In addition, it is necessary to identify potential stress-responsive allelic variations through dissecting the mechanisms underpinning transcriptional regulation and functional differentiation of duplicated genes in crops. Transgenic methods have been used to introduce superior genes into crops which confer high-yield potential but are accompanied by poor stress resistance. However, stress-tolerant phenotypes may be engineered through identifying differences in copy number, gene structure, expression level, and protein coding sequences of stress-responsive genes between tolerant and sensitive accessions. Despite the potential benefits of having several copies of stress response genes in the genome, genetic redundancy can also confound screening for distinct phenotypes. Gene-centric approaches, such as those utilizing CRISPR-Cas systems, could be used to achieve targeted improvements, reducing time and potentially increasing success rates. These approaches have been successful in model plants as well as in polyploid wheat, enabling functional genetic analysis in complex polyploid genomic backgrounds [225,226].

In summary, with natural selection continuing to serve as the cornerstone of evolutionary theory, we must expedite a deeper understanding of the origins, evolutionary mechanisms, and genetic basis of domestication that have culminated in modern wheat (Figure 2). Utilizing the genetic diversity and exploiting differences in wheat and its wild relatives is the foundation of germplasm innovation for the foreseeable future. At the same time, the threats posed by the increasing frequency and severity of global climate change warrant ongoing, extensive study of the genetic basis of traits that enable crop adaptation to environmental change. This advanced understanding will provide sound theoretical support for developing high-performance, stress-resilient germplasm and genetic resources for molecular breeding. Finally, the phenotypic characterization of alleles that enable continued high crop performance under variable or extreme field conditions is also essential for the effective deployment of stress-responsive genes identified in the laboratory.

## Figures and Tables

**Figure 1 plants-12-02465-f001:**
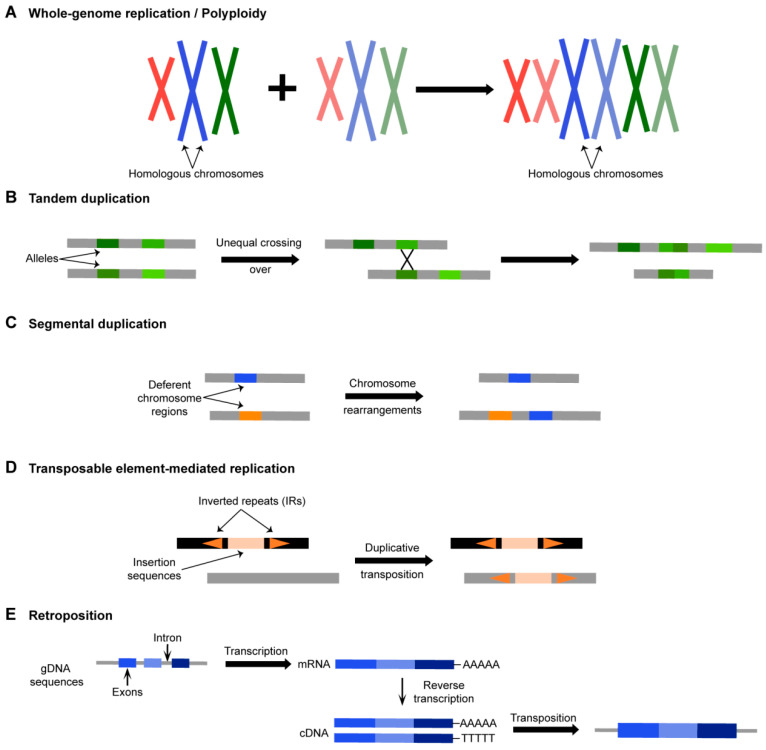
Mechanisms of gene duplication. (**A**) Whole-genome duplication (WGD) or polyploidy to produce duplicate genes. (**B**) Tandem duplication: duplication of a gene via unequal crossing-over between similar alleles. A chromosomal region within 200-kb containing two or more genes is defined as a tandem duplication event. (**C**) Segmental duplication: multiple genes through polyploidy followed by chromosome rearrangements. (**D**) Transposon-mediated duplication. (**E**) Retroduplication. mRNA, which has been transcribed and cleaved, goes through a reverse transcription process to form cDNA, which is then randomly inserted into a chromosome to form a new duplication gene.

**Figure 2 plants-12-02465-f002:**
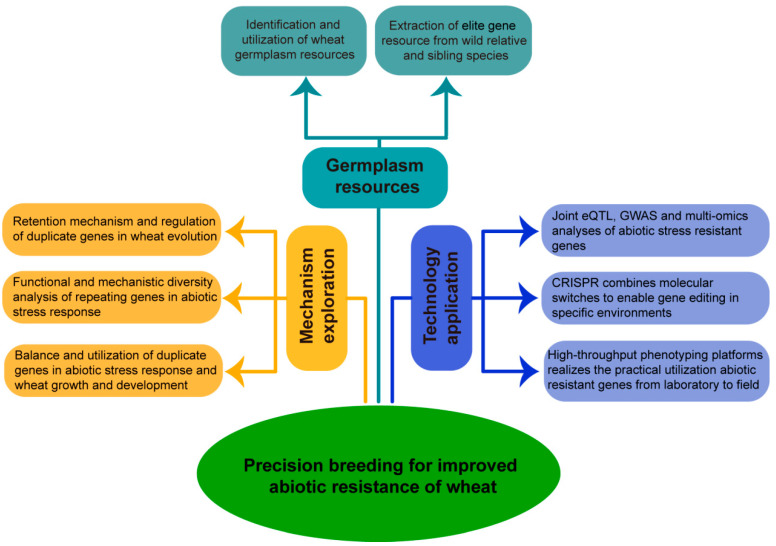
Prospects of abiotic stress-related duplicates in wheat. Making full use of the rich germplasm resources of the *Triticeae*, identification of abiotic stress-resistant varieties of bread wheat and the mining of superior genes in ancestral wild-type and closely related species. Explore the replication, retention, and evolutionary mechanisms of repetitive genes in wheat, as well as the diversity and differentiation of the functions and mechanisms of action of these genes in the abiotic response process, and rationalize the use of repetitive genes to balance their effects in stress resistance and wheat yield to achieve the desired results of stress resistance and yield. Make full use of multi-omics analysis platforms such as bioinformatics analysis and phenome in pursuit of application of excellent genes from greenhouse to field production. This proposal provides a potential precision-breeding method for improved abiotic resistance of wheat.

**Table 1 plants-12-02465-t001:** Recent examples of duplicate genes in wheat.

	Gene Name	Gene Product	Environmental Condition	Reference
Transcription factor	*TaSNAC8-6A*	NAC transcription factors	drought stress	[71]
	*TaNAC071-A*	NAC transcription factors	drought stress	[72]
	*TaSNAC4-3D*	NAC transcription factors	drought stress	[73]
	*TaSNAC4-3A*	NAC transcription factors	drought stress	[74]
	*TaDTG6-B*	Dehydration-responsive element-binding protein	drought stress	[75]
	*TaHsfA6f*	Heat shock factor	heat stress	[76]
	*TaHsfA2e-5D*	Heat shock transcription factor	drought and heat stress	[77]
	*TaHsfA6b-4D*	Heat shock transcription factor	heat stress	[78]
	*TaHsfC2a*	Heat shock factor	heat stress	[79]
	*TaMYB344*	MYB transcription factors	drought, heat, and salt stress	[80]
	*TaMYB56-B*	MYB transcription factors	freezing and salt stress	[81]
	*TabZIP60*	Basic leucine zipper proteins	heat stress	[82]
	*TabZIP14-B*	bZIP transcription factors	salt and freezing stress	[83]
	*TaWRKY75-A*	WRKY domain protein	salt stress	[84]
	*TaWRKY1-2D*	WRKY transcription factors	drought stress	[85]
	*TaERF-6-3A*	AP2/ERF transcription factors	drought and salt stress	[86]
	*TaGRF6-A*	General regulatory factors	salt stress	[87]
	*TaDrAp1, TaDrAp2*	Down-regulator associated protein	drought stress	[101]
	*TaFDL2-1A*	bZIP transcription factor	drought stress	[102]
	*TaMpc1-D4*	MYB transcription factors	drought stress	[103]
	*TaGT2L1D*	trihelix transcription factors	drought stress	[104]
	*TaNFYC-A7*	Recombinant Nuclear Transcription Factor	drought stress	[105]
	*TaAIDFα*	CRT/DRE-binding factor	cold stress	[107]
	*TaAREB3*	ABA-responsive element-binding proteins	cold stress	[108]
	*TaWD40-4B.1*	WD40 transcription factors	drought stress	[151]
	*TaRAV4 and TaRAV5*	RAV (related to ABI3/VP1) transcription factor	drought stress	[161]
	*TaZHD1 and TaZHD10*	Zinc finger homeodomain class transcription factors	drought stress	[166]
	*TaCBF14 and TaCBF15*	C-repeat/DREB binding factors	cold stress	[167]
	*TaBTF3*	Basic transcription factor 3	cold stress	[168]
	*TaRN2*	ASYMMETRIC LEAVES2 (AS2)/LATERAL ORGAN BOUNDARIES (LOB) domain transcription factor	heat stress	[169]
	*TaOPF29a*	OVATE family proteins	drought stress	[170]
	*TtNTL3A*	NAC transcription factors	drought and salt stress	[171]
Cytoprotective protein/enzyme	*TaCAT3*	Catalase	cold stress	[49,50]
	*TaSOD2*	Superoxide dismutases	salt stress	[51,52,53]
	*TaAPX-R*	Ascorbate peroxidase	drought and salt stress	[54]
	*TaGPX*	Glutathione peroxidase genes	salt stress	[55,56]
	*TaPRX-2A*	Peroxidase gene family	salt stress	[57]
	*TaGR2-B1*	Glutathione reductase	salt stress	[58]
	*TaADF16*	Actin depolymerizing factor	cold stress	[100]
	*TaHVA1*	Group 3 Late Embryogenesis Abundant protein	drought and heat stress	[160]
	*TaEXPA8*	Expansin protein	cold stress	[163]
Transporters	*TaOSCAs*	Hyperosmolality-gated calcium-permeable channels	drought, salt, heat stress	[62,63]
	*TaBORs*	BOR transporter family	drought, salt, heat stress	[64]
	*TaMSL*	Mechanosensitive channel of small conductance-like	drought, salt, heat stress	[66]
	*TaCaCA*	Ca2+/cation antiporters	drought, salt, heat stress	[67]
	*TaNHX4-B.1 and TaNHX4-B.4*	Cation proton antiporter	drought, salt, heat stress	[68]
	*TaACAs and TaECAs*	P-type II Ca2+ATPases	drought, salt, heat stress	[69]
	*TaSOS1*	Na+/H+ antiporter	salt stress	[146,147]
	*TaHKT1;5-D, TmHKT1;5- A*	Na+ transporter	salt stress	[172,173]
	*HKT1;4*	Na⁺ transporter	salt stress	[174]
	*TaCLC; TaCCC*	Chloride channel; cation chloride co-transporter	salt stress	[175]
	*TdHKT1;4*	Na⁺ transporter	salt stress	[176]
Homeobox genes	TaHD-Zip	HD-Zip gene family	salt and drought stress	[88]
	*TaKNOX11-A*	TALE superfamily protein	drought, salt stress	[89,90]
	*TaZF-HD*	Zinc Finger-Homeodomain Transcriptional Factors	drought, salt, and cold stress	[91]
	TaWUS and TaWOX14	WUSCHEL-Related Homeobox	drought, salt, heat stress	[92,93]
	TaPHD	Plant homeodomain (PHD) transcription factors	cold, drought, and heat stress	[177]
Metabolism-related enzyme	*TaLTPIb.1, TaLTPIb.5, and TaLTPId*	Non-specific lipid transfer proteins	cold stress	[97]
	*TaCYP81D5*	Cytochrome P450 protein	salt stress	[120]
	*TaHSP70s*	Heat shock protein	heat stress	[141]
	*TraeALDH7B1-5A*	Aldehyde dehydrogenase	drought stress	[144]
	*TaHXK3-2A*	Hexokinase	drought stress	[178]
	*TaTPS11*	Trehalose 6-phosphate synthase	cold stress	[179]
	*TaG6PDH*	Glucose-6-phosphate dehydrogenase	cold stress	[180]
	*TaHSP90s*	Heat shock protein	heat stress	[181]
	*TaFER-5B*	Ferritin	heat stress	[182]
	*TaDEAD-box57-3B*	DEAD-box RNA Helicase	drought and salt stress	[183]
	*TaCER1-6A,TaCER1-1A*	Alkane biosynthesis gene	drought stress	[184,185]
Cell signaling protein/enzymes	*TaCPK7 and TaCPK12*	Calcium-dependent protein kinases	drought stress	[139]
	*TaRN1*	Serine/threonine protein kinase	salt stress	[169]
	*TaPYL1*	ABA receptor	drought stress	[186]
	*CYCB2, CDKA1*	B2-type cyclin in mitotic; cyclin-dependent kinases	drought stress	[187]
	*TaSCPL184-6D*	Serine carboxypeptidase-like protein	salt stress	[188]
Receptor like protein kinase	*TaLRRKs*	Leucine-rich repeat kinase	heat and drought, and salt	[42,43]
	*TaCRK68-A*	Cysteine-rich receptor-like kinases	heat, drought, cold and salt stress	[44]
	*TaTLPs*	Thaumatin-like protein kinases	heat and drought, and salt	[45]
	*TaPERKs*	Proline-Rich Extensin-like Receptor Kinases	heat stress	[46,47]
Epigenetic regulation genes	*TaMBD2*	Methyl CpG-binding domain proteins	cold stress	[94]
	*TaHAG1*	Histone acetyltransferase	heat and salt stress	[164,165]
	*TaCMT*	Cytosine-5 DNA methyltransferases	drought, heat stress	[189]
	*Tr-7A-JMJ1,Tr-1B-JMJ3*	Histone demethylase	drought stress	[190]
	*TaSIRFP-3A,TaSIRFP-3B*	RING-HC-type E3 ligases	cold stress	[191]
	*TaPUB2/TaPUB3*	U-box E3 ubiquitin ligase	drought stress	[192]
Other stress response genes	TaCHYR2.1, TaCHYR9.2, TaCHYR11.1	CHY zinc-finger and RING finger protein	salt stress	[129]
	*TaICE41 and TaICE87*	Inducer of CBF expression	cold stress	[162]
	*Vrn-B1, Vrn-D3*	Vernalization genes	drought and heat stress	[193,194]
	*TaBI-1.1*	Bax Inhibitor	heat stress	[195]

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
