# Peer review of "Duplicate Genes Contribute to Variability in Abiotic Stress Resistance in Allopolyploid Wheat"

_plants, 2023, doi:10.3390/plants12132465_

Round 1

Reviewer 1 Report

This review summarizes the current knowledge of gene duplication in wheat, including the principle of gene duplication and its effects on gene function, the diversity of duplicated genes, and how they have functionally diverged. The authors also explain how duplicated genes contribute to abiotic stress response and the mechanisms of duplication. The draft describes the prospects and direction of future efforts in the short term regarding the elucidation of replication and retention mechanisms of repetitive genes related to abiotic stress response in wheat, excellent gene function research, and practical applications.

The article is well-written, interesting, and useful as a unified and systematized source of information on what is known so far on the subject. It is an undeniable fact that severe weather events or weather conditions are becoming more frequent due to global climate change, and indeed, today the world is facing the prospect of food insecurity. Figure 2 reflects the outlook for abiotic stress-related duplicates in wheat and is an appropriate summarizing conclusion to the review.

There are 175 references covering the period from 1985 to 2023. The scientific problem is comprehensively addressed from different scientific perspectives.

Some remarks:

  • Latin names must be in italics (lines 40-41).

  • Line 130: The word "respectively" is of a different size than the rest of the text.

  • Lines 225-230: The text is in a different size than the rest of the text.

The language in which the article is written is good. The correct terms are used; it only requires minor grammar and punctuation corrections.

Author Response

Reviewer #1: 

This review summarizes the current knowledge of gene duplication in wheat, including the principle of gene duplication and its effects on gene function, the diversity of duplicated genes, and how they have functionally diverged. The authors also explain how duplicated genes contribute to abiotic stress response and the mechanisms of duplication. The draft describes the prospects and direction of future efforts in the short term regarding the elucidation of replication and retention mechanisms of repetitive genes related to abiotic stress response in wheat, excellent gene function research, and practical applications.

The article is well-written, interesting, and useful as a unified and systematized source of information on what is known so far on the subject. It is an undeniable fact that severe weather events or weather conditions are becoming more frequent due to global climate change, and indeed, today the world is facing the prospect of food insecurity. Figure 2 reflects the outlook for abiotic stress-related duplicates in wheat and is an appropriate summarizing conclusion to the review.

There are 175 references covering the period from 1985 to 2023. The scientific problem is comprehensively addressed from different scientific perspectives.

Response: The authors would like to thank the reviewer for their time and careful attention to detail in helping us to improve the quality of our analyses, and consequently, the rigor of our conclusions. We have done our best to address each of the reviewer’s questions and concerns, and revised the text as necessary. Hopefully the reviewer finds that our revised manuscript now meets the high standards necessary for publication in Plants, and we again thank them for their invaluable assistance.

Some remarks:

  1. Latin names must be in italics (lines 40-41).

Response: We are grateful for the reviewer's careful attention to detail. We have modified Latin name to italic.

“phenotypic changes in flowering time or plant height were detected in several domesticated crops, including wheat (Triticum aestivum), cotton (Gossypium hirsutum), and soybean (Glycine max).”

  1. Line 130: The word "respectively" is of a different size than the rest of the text.

Response: We appreciate the reviewer’s careful attention to the detail. according to the reviewers' comments.

“For example, overexpressing TaHsaA2d or TaHsaA6f in wheat not only increases heat tolerance, but also drought and salt stress, respectively.”

  1. Lines 225-230: The text is in a different size than the rest of the text.

Response: The authors would like to thank the reviewer for their time and careful attention to detail in helping us to improve the quality of our reviews. We have corrected the text size consistently according to the reviewers' comments.

“Another recent study revealed that TaWD40-4B.1, located in the main drought tolerance QTL, qDT4B, along with an early termination codon generated by a nonsense mutation in TaWD40-4B, was highly correlated with drought resistance in a natural wheat population. These findings showed that wheat carrying TaWD40-4B.1C

had significantly higher drought resistance than wheat carrying TaWD40-4B.1T [97].”

Reviewer 2 Report

In the review article, authors discuss the role of duplicated genes in allopolyploid wheat. Article is interesting but lack several updated information about the duplicated genes reported in last few years in wheat. They should include the information about all the duplicated genes involved in stress response in wheat. This review seems to be very limited to a few gene family only

1.      Receptor like kinases are an important class in stress response, which has been completely ignored. I could see numerous studies related to RLKs showing paralogous =genes info. For ex, Lectin receptor kinases,  leucine rich repeat receptor like kinases (TaLRRKs), Cysteine-rich receptor-like kinases , LysM domain-containing proteins , thaumatin-like proteins and thaumatin-like kinases, etc in allohexaploid wheat. These are al recently reported .

2.      Similarly, enzymatic anti-oxidants are important stress responsive class of proteins that have been ignored, Several studies reported duplication in these genes. Authors may see and include that info regarding SOD, APX, GPX, GR, etc. in bread wheat.

3.      Simillarly, various transporters responding to numerous stress are missing that have been studied well in bread wheat, including OSCA Genes in Bread Wheat, Bor genes, Piezo , mechanosensitive channel of small conductance-like (MSL) genes , CaCA, Ca ATPases, CPA etc. These are all related to the transport of Ca, Na and other important molecules during stress responsponce.

4.        Besides a few Homeobox genes like, WOX, TALE etc have been well discussed in the recent years.

5.      The studies might have described the expression data or functional data for duplicated genes but that should be the part of review.

6.      This review should be properly revised with all the regarding duplicated genes involved or responding to abiotic stress in allopolyploid wheat as represented in the title. They may discuss different families sequentially.  Table 1 should be revised properly or may be split in gene family wise. 

Needs significant improvement

Author Response

Reviewer #2: 

In the review article, authors discuss the role of duplicated genes in allopolyploid wheat. Article is interesting but lack several updated information about the duplicated genes reported in last few years in wheat. They should include the information about all the duplicated genes involved in stress response in wheat. This review seems to be very limited to a few gene family only.

  1. Receptor like kinases are an important class in stress response, which has been completely ignored. I could see numerous studies related to RLKs showing paralogous =genes info. For ex, Lectin receptor kinases, leucine rich repeat receptor like kinases (TaLRRKs), Cysteine-rich receptor-like kinases, LysM domain-containing proteins , thaumatin-like proteins and thaumatin-like kinases, etc in allohexaploid wheat. These are all recently reported.

Response: The authors would like to thank the reviewers for their time and careful attention to the results, which helped us to improve the quality of our paper. As the reviewers reminded, the wheat-like receptor protein kinase superfamily have been identified and their repeat events and their similarities and differences in expression data have been investigated in recent years. We have carefully consulted several updated information about the duplicated genes reported in last few years in wheat, and added this section to the text (Lines 128-140) and to Table 1 as appropriate.

Receptor-like protein kinases (RLKs), the largest gene family in plants, play critical roles in the regulation of plant developmental processes, signaling transmission and stress resistance[38-40].. In recent years, the RLK family of wheat has been identified, and collinearity events and tandem gene clusters results suggested that polyploidization and tandem duplication events contributed to the RLK member expansions of T. aestivum [38]. Among them,lectin receptor kinases (LEC-RLK) [41], leucine rich repeat receptor like kinases (LRR-RLKs) [42, 43], cysteine-rich receptor-like kinases (CRKs) [44], thaumatin-like proteins and thaumatin-like kinases (TLPs) [45], and proline-rich extensin-like receptor protein kinases (PERKs) [46, 47]. Gene replication events were identified in these families, and some of these replication gene pairs had differential expression data under abiotic stress, such as. TaTLP14-A1/B1/B5 were up-regulated in expression under osmotic stress, while TaTLP14-B4 were the opposite [45]

  1. Similarly, enzymatic anti-oxidants are important stress responsive class of proteins that have been ignored, Several studies reported duplication in these genes. Authors may see and include that info regarding SOD, APX, GPX, GR, etc. in bread wheat.

Response: We thank the reviewer for their constructive comments and useful suggestions, the latter of which have helped us improve the manuscript. Lines 140-151

Enzymatic anti-oxidants are important stress responsive class of proteins that scavenging excess reactive oxygen specie (ROS) in the presence of cofactors such as copper and zinc ions [48]. Several studies identified and reported duplication in antioxidant genes, including catalase (CAT) [49], superoxide dismutase (SOD) [50-51], ascorbate peroxidase (APX) [52], glutathione peroxidase (GPX) [53], peroxidase (POD) [54], and glutathione reductase (GR) genes in bread wheat [55]. For example, TaCAT3-A1 and TaCAT3-A2 were found to be clustered into tandem duplication event regions, while the number of cis-elements in the promoter of TaCAT3-A2 was more than TaCAT3-A1; moreover, TaCAT3-A1/A2 contained cis-elements associated with cold response, but not exist in other subgenome copies TaCAT3-B and TaCAT3-D [49], which suggesting that there may be differences in cold response of TaCAT3 homologous gene groups.

  1. Simillarly, various transporters responding to numerous stress are missing that have been studied well in bread wheat, including OSCA Genes in Bread Wheat, Bor genes, Piezo , mechanosensitive channel of small conductance-like (MSL) genes , CaCA, Ca ATPases, CPA etc. These are all related to the transport of Ca, Na and other important molecules during stress responsponce.

Response: According to the reviewer's suggestion, we transporters responding to abiotic stress. This information has been added to the revised manuscript on Lines 152-164 and Table 1.

In plants, various transporters were reported related to the transport of Ca, Na and other important molecules during stress responsponce to maintain ion homeostasis in the plant cell [56-58]. Hyperosmolality-gated calcium-permeable channels (OSCA) [59, 60], Boron transporters (BOR) [61, 62],,mechanosensitive channel of small conductance-like (MSL) genes [63], and Ca2+/cation antiporters (CaCA) in bread wheat [64] gene family were identified and analysis of their expression pattern under abiotic stress. For example, the paraline homologous genes TaMSL4-A1 and TaMSL7-A in MSL were both drought-induced, but the expression of TaMSL4-A1 was inhibited under heat stress and in the early stage of salt stress; while the expression of TaMSL7-A was up-regulated under heat stress and the under early salt stress, which indicating the response of paraline homologous genes was different under different abiotic stresses, and the expression was also diverse and began to express and function under different levels of stress [63].

  1. Besides a few Homeobox genes like, WOX, TALE etc have been well discussed in the recent years. The studies might have described the expression data or functional data for duplicated genes but that should be the part of review.

Response: According to the reviewer's suggestion, The information have been added to Lines 164-169 and Table 1.

A similar phenomenon occurs with transcription factor, in which orthologous or paralogue genes that are produced by gene duplication exhibit identical or opposite expression patterns when subjected to abiotic stress, including NAC [65-68], DREB [69], Hsf [70-73], MYB [74, 75], bZIP [76, 77], WRKY [78, 79], AP2/ERF [80], GRF [81], and the homeobox genes HD-Zip [82], TALE [83], ZF-HD [84], and WOX [85, 86].

  1. This review should be properly revised with all the regarding duplicated genes involved or responding to abiotic stress in allopolyploid wheat as represented in the title. They may discuss different families sequentially. Table 1 should be revised properly or may be split in gene family wise.

Response: We appreciate the reviewer's suggestion. We have carefully revised the manuscript, improved and added related duplicated genes involved or responding to abiotic stress in allopolyploid whea according to the reviewer's comments, and classified these genes according to different biological pathways in Table 1. Finaly, we checked the references carefully for accuracy.

Round 2

Reviewer 2 Report

The Ms has been revised properly, It may be accepted with some other minor changes.

Authors have revised Ms very carefully with numerous new info regarding the role of various duplicated genes including RLKs, TFs, transporters etc. Still a few have been left, which would make this review more inclusive.

Cation-Proton antiporters,   P-type II Ca2+ATPases ,  LysM domain-containing proteins  ,   thaumatin-like protein kinase are other classes well studied for duplicated gene function in wheat in recent years, that can also be discussed in respective paragraphs.    

Ref 43 seems to be wrongly mentioned in the reference list. It should be for GPX; Gene architecture and expression analyses provide insights into the role of glutathione peroxidases (GPXs) in bread wheat.  The duplication of TaTALE including KNOX and BLH family has been reported in detail here www.mdpi.com/2223-7747/11/5/587

In the case of SOD and Catalase genes in bread wheat, I could see that the first reports in wheat were published in Agri-gene and J hazardous material, that may also be the part of discussion. The quoted references by Jiang and Zhang et al. are repeated reports from the earlier studies. 

The revised sections needs to be highlighted with colour changes or may be done in track changes. The paragraphs in Ms is also not properly aligned.

It seems fine.

Author Response

Reviewer #2:

The Ms has been revised properly, It may be accepted with some other minor changes.

Authors have revised Ms very carefully with numerous new info regarding the role of various duplicated genes including RLKs, TFs, transporters etc. Still a few have been left, which would make this review more inclusive.

  1. Cation-Proton antiporters, P-type II Ca2+ ATPases, LysM domain-containing proteins, thaumatin-like protein kinase are other classes well studied for duplicated gene function in wheat in recent years, that can also be discussed in respective paragraphs.

Response: We thank the reviewer for their constructive comments and useful suggestions, the latter of which have helped us improve the manuscript. Cation-Proton antiporters, P-type II Ca2+ ATPases, LysM domain-containing proteins, thaumatin-like protein kinase has been discussed and discussed in Lines 154-168 and Table 1.

Hyperosmolality-gated calcium-permeable channels (OSCA) [62, 63], Boron transporters (BOR) [64, 65], mechanosensitive channel of small conductance-like (MSL) genes [66], Ca2+/cation antiporters (CaCA) [67], cation proton antiporter (CPA) [68], P-type II Ca2+ ATPases [69], and thaumatin-like protein kinases (TLPKs) in bread wheat gene family were identified and analysis of their expression pattern under abiotic stress [70]. For example, the genes pairs TaMSL4-A1 and TaMSL7-A in MSL were both drought-induced, but the expression of TaMSL4-A1 was inhibited under heat stress and in the early stage of salt stress; while the expression of TaMSL7-A was up-regulated under heat stress and the under early salt stress, which indicating the response of paraline homologous genes was different under different abiotic stresses, and the expression was also diverse and began to express and function under different levels of stress [66]. The cation proton antiporter (CPA) superfamily, including K+ efflux antiporter (KEA), and cation/H+ exchanger (CHX) family proteins, and Na+/H+ exchanger (NHX), TaNHX4-B.1 and TaNHX4-B.4 facilitated differential drought, salt and heat stress tolerance to Escherichia coli [68].

  1. Ref 43 seems to be wrongly mentioned in the reference list. It should be for GPX; Gene architecture and expression analyses provide insights into the role of glutathione peroxidases (GPXs) in bread wheat. The duplication of TaTALE including KNOX and BLH family has been reported in detail here mdpi.com/2223-7747/11/5/587.

Response: The author is very patient and careful of reviewers, which is very helpful for us to improve the article. Ref 43 is indeed cited for leucine-rich repeat receptor-like kinases (LRRKs) rather than GPX. The references cited by GPX in text was Ref 55 in the reference list.

  1. Tyagi, S., Himani., Sembi. JK., Upadhyay, SK. Gene architecture and expression analyses provide insights into the role of glutathione peroxidases (GPXs) in bread wheat (Triticum aestivum ). J Plant Physiol. 2018, 223:19-31.

According to the reviewer's suggestion, we have added the duplication of Homeobox genes TaTALE family reported by Rathour et al. The have been revised in Lines 173-18 and Table 1.

The homeobox genes HD-Zip [88], TALE [89, 90], ZF-HD [91], and WOX [92, 93]. Taking TaTAIL as an example, Rathour et al systemically identified and analyzed TAIL family members in wheat, including gene and protein structural properties, phylogeny, expression patterns. The gene duplication events were identified, a gene pairs TaTALE8-4A3 and TaTALE8-4A1 produced by fragment replication and five tandem duplicate gene pairs such as TaTALE1-1A2 and TaTALE1-1A1. The cis-acting elements of these repeating gene promoters and their expression data were different under heat and salt stress, indicating that the response of repeating genes to abiotic stress was diverse [89].

  1. In the case of SOD and Catalase genes in bread wheat, I could see that the first reports in wheat were published in Agri-gene and J hazardous material, that may also be the part of discussion. The quoted references by Jiang and Zhang et al. are repeated reports from the earlier studies.

Response: According to the reviewer's suggestion, first reports case of SOD and Catalase genes in bread wheat were review to discussion and added to Lines 142-146 and Table 1.

Several studies identified and reported duplication in antioxidant genes, including catalase (CAT) [49, 50], superoxide dismutase (SOD) [51-53], ascorbate peroxidase (APX) [52], glutathione peroxidase (GPX) [55], peroxidase (POD) [56], and glutathione reductase (GR) genes in bread wheat [57].

  1. The revised sections needs to be highlighted with colour changes or may be done in track changes. The paragraphs in Ms is also not properly aligned.

Response: According to the reviewer's suggestion, revised sections had marked in red colour. The paragraphs in Ms was also properly aligned in word. The references revised were checked carefully for accuracy.

Round 3

Reviewer 2 Report

The Ms may now be accepted.